# Impact of COVID-19 on the mental health of public university hospital workers in Brazil: A cohort-based analysis of 32,691 workers

Adriana Ferreira Barros-Areal[1,2☯], Cleandro Pires Albuquerque[1,3☯], Nayane Miranda Silva[1,3‡], Rebeca da Nóbrega Lucena Pinho[1,3‡], Andrea Pedrosa Ribeiro Alves Oliveira[4‡], Dayde Lane Mendonça da Silva[3,5‡], Ciro Martins Gomes[1,3,4,6‡], Fernando Araujo Rodrigues de Oliveira[3‡], Patrícia Shu Kurizky[1,3‡], Ana Paula Monteiro Gomides Reis[7‡], Luciano Talma Ferreira[3‡], Rivadávio Fernandes Batista de Amorim[1‡], Marta Pinheiro Lima[8‡], Claudia Siqueira Besch[8‡], Giuseppe Cesare Gatto[8‡], Thais Ferreira Costa[2‡], Everton Nunes da Silva[9‡], Heidi Luise Schulte[1‡], Laila Salmen Espindola[1‡], Licia Maria Henrique da Mota[1,3☯]*

1 Programa de Pós-Graduação em Ciências Médicas, Faculdade de Medicina, Universidade de Brasília, Brasília, Brazil, 2 Secretaria de Estado de Saúde do Distrito Federal–SES DF, Brasília, Brazil, 3 Hospital Universitário de Brasília, Universidade de Brasília, Brasília, Brazil, 4 Faculdade de Medicina, Universidade de Brasília, Brasília, Brazil, 5 Faculdade de Ciências da Saúde, Universidade de Brasília, Brasília, Brazil, 6 Núcleo de Medicina Tropical, Faculdade de Medicina, Universidade de Brasília, Brasília, Brazil, 7 Centro Universitário de Brasília (Uniceub), Brasília, Brazil, 8 Empresa Brasileira Serviços Hospitalares- EBSERH, Brasília, Brazil, 9 Faculdade de Ceilândia, Universidade de Brasília, Brasília, Brazil

☯ These authors contributed equally to this work.
‡ NMS, RNLP, APRAO, DLMS, CMG, FARO, PSK, APMGR, LTF, RFBA, MPL, CSB, GCG, TFC, ENS, HLS and LSE also contributed equally to this work.
* artigos.ppgcm@gmail.com

**Data Availability Statement:** All relevant data are within the manuscript and its Supporting Information files.

## Abstract

### Background

In early 2020, the COVID-19 pandemic paralyzed the world and exposed the fragility of health systems in the face of mass illness. Health professionals became protagonists, fulfilling their mission at the risk of physical and mental illness. The study aimed to evaluate absenteeism indirectly related to SARS-CoV-2 infection in a large population of health care professionals.

### Methods

An observational longitudinal repeated measures study was performed, including workers linked to 40 public university hospitals in Brazil. All causes of absenteeism were analyzed, focusing on those not directly attributed to COVID-19. Results for the same population were compared over two equivalent time intervals: prepandemic and during the pandemic.

### Findings

A total of 32,691 workers were included in the study, with health professionals comprising 82.5% of the sample. Comparison of the periods before and during the pandemic showed a 26.6% reduction in work absence for all causes, except for COVID-19 and mental health-

**Funding:** This study was supported: Conselho Nacional de Desenvolvimento Científico e Tecnológico (CNPq), Coordenação de Aperfeiçoamento de Pessoal de Nível Superior (CAPES), Universidade de Brasília (UnB – DPG 0004/2021) and ArboControl Project (TED 74/2016).

**Competing interests:** The authors have declared that no competing interests exist.

related absence. Concerning work absence related to mental health, the odds ratio was 39.0% higher during the pandemic. At the onset of the pandemic, there was an increase in absenteeism (all causes), followed by a progressive reduction until the end of the observation period.

## Interpretation

Work absence related to mental illness among health care professionals increased during the COVID-19 pandemic, highlighting the need for health care managers to prioritize and implement support strategies to minimize absenteeism.

## Introduction

On March 11, 2020, the World Health Organization declared that the global health emergency triggered by the emergent coronavirus—SARS-CoV-2, the etiological agent of COVID-19, was characterized as a pandemic [1]. Initial reports of the disease emerged in late 2019, with reports of successive cases on all continents and dramatic health and social impacts [2, 3]. One year later, the world had accumulated more than 116 million cases and suffered approximately 2,5 million deaths [4]. In March 2021, Brazil was one of the three countries with the highest number of cases, reporting 10,8 million confirmed cases and >262,000 deaths, surpassed only by the United States of America and India [4, 5].

The Brazilian public health system was already overwhelmed by the continuous demand to treat endemic infectious diseases and a high prevalence of chronic noncommunicable diseases [6]. In pronounced social inequality, most of the population depends exclusively on the Unified Health System (SUS), a public service that suffers from chronic financial shortages and excessive demand [6, 7].

The COVID-19 pandemic exposed health systems to an unprecedented challenge, strongly impacting frontline health workers. In March 2020, there were reports of >5000 infected health professionals in Italy, including doctors, nurses and technicians [8]. The removal of infected professionals from the workplace overloaded colleagues with patients and caused them more stress which triggered an increase in burnout syndrome and other types of mental illness [9]. Thus, caring for these professionals' physical and mental health has become a strategic issue for maintaining the workforce during the pandemic [10, 11]. A systematic review of 117 studies published in August 2020 assessed the impact of health emergencies and epidemics on the mental health of health professionals, revealing a higher prevalence of mental illness [12].

Hence, it is necessary to assess the impact of health professional sick leave during the COVID-19 pandemic. Discussions of many aspects of this topic remain sparse in the literature, particularly in developing countries such as Brazil which have historically suffered from health professional shortages and lack of funding [13].

The objective of the present study was to investigate the indirect impact of COVID-19 on the health system workforce by assessing absenteeism from all causes, not directly attributable to suspected or confirmed SARS-CoV-2 infection, with a particular focus on mental health-related absenteeism.

## Methods

### Ethical aspects

The research was approved by the National Research Ethics Commission—CEP / CONEP under registration number CAAE: 31785720700005558 and adopted by opinion substantiated number 4,054,379.

### Study context

Brazil has a public health system with free universal access to primary, medium- and high- complexity care for the entire population [7]. The country has 50 public university hospitals which are important training centers for human resources which provide both medium- and high-complexity care for the Brazilian population. The Brazilian Company of Hospital Services (EBSERH) manages 40 public hospitals distributed nationwide and employs approximately 60 thousand professionals, with more than 32,000 permanent workers hired directly by EBSERH. The other 27,000 workers have temporary contracts or were transferred from other institutions. This study analyzed the leave records for permanent workers maintained in the EBSERH database.

### Type of study and design

This is an observational longitudinal repeated measures study. Work leave was assessed in two different periods: (i) prepandemic, 03/01/2019 to 07/31/2019, and (ii) during the COVID-19 pandemic, 03/01/2020 to 07/31/2020. The study population was comprised of health professionals and support staff for health care activities linked to the EBSERH network in the two periods of interest.

### Characterization of absenteeism and data collection

Data regarding work absences taken by health professionals and support staff were obtained from the administrative human resources information databases. The definition of a work absence is a period in which the employee did not work, measured in days and classified according to the reason for the leave: health-related, administrative, family-related (marriage, paternity, maternity or adoption leave), illness or death of a family member, blood donation, abortion and its complications, or occupational accidents. Due to suspicion or confirmation of SARS-CoV-2 infection, periods when employees switched to teleworking were not counted among the outcomes of interest in the study. In addition to data relating to the classification of work absences, we collected variables such as specific causes, the duration of the absence, and sociodemographic variables.

### Inclusion and exclusion criteria

Healthcare workers from all professional categories were included in the study, including those who maintained direct contact with patients, such as doctors, nurses, nursing technicians and physiotherapists, as well as workers supporting health activities without direct contact with patients such as administrative and support professionals. Workers linked to EBSERH in both periods of interest were included, while workers linked to EBSERH during only one of the observation periods were excluded.

### Statistical analysis

We compared the pre- and during-pandemic periods in terms of the occurrence of work absences and their causes using statistical methods for the analysis of repeated measurements.

The proportions of individuals who had work absences due to several causes unrelated to suspected or confirmed COVID-19 infection, including mental disorders, were assessed using the McNemar test. Changes in the number of events (counts) of work absences due to mental illness per individual were assessed using a generalized estimating equation (GEE) model based on the negative binomial distribution with a log-link function (unstructured correlation matrix). GEE models based on the binomial distribution with a logit-link function (unstructured correlation matrix) were used to assess differences between the genders regarding the proportions of individuals who had work absences due to mental illness and other causes. Using a mixed effects model (time as fixed effects; random intercepts), we assessed changes in the mean duration (days) of work absences due to mental diseases. Survival analyses with Kaplan-Meier curves and log-rank tests were used to assess changes in the time-to-event (of work absences) profiles across the observation periods and graphically evaluate the uniformity of such profiles within each period. Correlation between the number of confirmed COVID-19 cases in the city (per 100,000 habitants) and the number of work absence events due to all causes unrelated to suspected or confirmed COVID-19 infection (per 100 health care professionals) was assessed by Spearman's rho. P value were deemed significant if they were <0,05. All analyses were conducted in SPSS 25.

## Results

A total of 32,691 workers were included in the national study. Study participants from 36 university hospitals and four maternity schools in all five regions of the country were included in the study: 15,942 (48.8%) from the Northeast, 6130 (18.8%) from the Southeast, 4626 (14.2%) from the South, 4504 (13.8%) from the Central-West and 1489 (4.6%) from the North region.

The average age of the workers was 39,2 years (SD 7,52). There was a predominance of females at 70.3% (n = 22,982). The sample comprised of 82.5% health professionals and 13.7% support professionals, while the remaining 3.8% with no information about professional category could be either health or support professionals. The most frequent occupations among health professionals in the sample were nursing technicians (31.4%), doctors (21.0%), nurses (17.0%), health technicians (3.3%) and physiotherapists (3.1%); in addition to other categories (pharmacists, psychologists, social workers, nutritionists, speech therapists, occupational therapists, dentists), that totaled 7.7%.

In the period from 03/01/2020 to 07/31/2020, 10,994 individuals (33.6% of the contingent workers) were dismissed due to suspicion or confirmation of COVID-19, corresponding to a total of 21,295 dismissal events (more than one event possible per individual), with a total cumulative loss of 127,551 working days. In addition, 6504 individuals (19.9%) switched to remote work, at some point, either due to presenting with risk factors for severe forms of COVID-19 or providing services that did not require their physical presence in the workplace (to increase social distancing), corresponding to a cumulative total of 349,016 remote working days. Absences resulting from causes related to COVID-19 were not counted among the outcomes of interest in the present study.

The absolute frequency of absences due to causes not directly associated with infection or suspicion of infection with SARS-CoV-2 in the pandemic period was decreased compared to that in prepandemic period. In the latter, there were 41,469 work absences, while during the pandemic, there were 29,217 work absences. The percentage of individuals on leave due to causes not directly associated with COVID-19 was 43.6% in the pandemic period versus 51.3% in the prepandemic period (OR—odds ratio– 0,73; 95% CI: 0, 71–0,76 p <0·0001).

Regarding the classification of work absences, those related to health were more frequent than administrative absences in both periods of interest. Both classes of work absence showed

a statistically significant reduction during the pandemic compared to the prepandemic period (Table 1). Notably, the proportion of all leave classes to total leave remained the same in the prepandemic and pandemic periods.

Despite the reduction observed for absenteeism in general (except for cases of COVID-19), divergent behavior was observed in relation to mental illness-related absences. The percentage of workers who left due to mental illness during the prepandemic period was 2.5%, versus 3.4% in the pandemic period (OR 1,39; 95% CI 1,26–1,52; p <0·0001). Fig 1 compares the observation periods for various causes of work absence and shows a 39% greater incidence of mental illness during the pandemic.

The number of work absences by disease groups in the prepandemic period and during the COVID-19 pandemic (Fig 2) highlights the increase in absenteeism related to mental illnesses.

The number of work absences due to mental illness per person also increased during the pandemic compared to the previous period [4.07 leaves per 100 professionals, 95% CI: 3.73–4.42 versus 3.05 leaves per 100 professionals, 95% CI 2.77–3.36; RR (relative risk) = 1.32, 95% CI: 1.19–1.45; p <0.001]. The number of work absences per person due to mental illness was lower among men than women [2.37 versus 5.24 leaves per 100 professionals; RR = 0.45, 95% CI: 0.39–0.53; p <0.001]. There was an increase in the average duration of work absence due to mental illness during the pandemic compared to the previous period (22.7 days [20.7–24.7] vs. 18.7 days [16.4–20.9], respectively; p = 0.001). There was no significant difference between the sexes in the average duration of work absence: the average for men was 22.0 days [18.9–25.2], the average for women was 19.3 days [18.0–20.7]; p = 0.118.

Analysis of the Kaplan-Meier curves showed that there was an increase in work absence at the beginning of the pandemic for all causes compared to the prepandemic period. However, during the course of the pandemic, there was a progressive reduction in work absence for causes not directly related to COVID-19, which became less frequent and persisted until the end of the observation period, compared to the prepandemic period (Fig 3A). This same pattern was observed in relation to work absence due to health-related causes (Fig 3B).

In the cities in which university hospitals included in the study, we observed a correlation between health professional work absence for all causes (except COVID-19) and the cumulative prevalence of cases of COVID-19 (R = 0.358, p = 0.038) (Fig 4). The higher the number of COVID-19 cases were in the respective cities, the higher the number of instances of work absence due to causes not directly attributed to SARS-CoV-2 infection.

**Table 1. Frequency of absenteeism of individuals as classes of work absence in the pre- and pandemic periods and the odds ratio between the analysed groups.**

| Classification of work absence | Pre pandemic 2019 n (%) | During pandemic 2020 n (%) | Odds Ratio (95% IC) | P value[b] |
|---|---|---|---|---|
| Health related[a] | 13264 (40.6) | 11149 (34.1) | 0.76 (0.73–0.78) | <0.001 |
| Administrative | 4340 (13.3) | 2858 (8.7) | 0.63 (0.59–0.66) | <0.001 |
| Marriage, paternity, maternity, adoption | 1473 (4.5) | 1348 (4.1) | 0.91(0.84–0.98) | <0.018 |
| Sickness or death of a family member | 911 (2.8) | 786 (2.4) | 0.86 (0.78–0.95) | <0,002 |
| Blood donation | 783 (2.3) | 515 (1.6) | 0.69 (0.61–0.78) | <0,0001 |
| Abortion and complications | 153 (0.81) | 118 (0.63) | 0.77 (0.60–0.98) | <0,033 |
| Occupational accidents | 88 (0.3) | 46 (0.1) | 0.52 (0.36–0.75) | <0,0001 |
| Total | 16774 (51.3) | 14261 (43.6) | 0.73 (0.71–0.76) | <0,0001 |

[a] Not directly related to SARS-CoV-2 infection.

[b] p value based on McNemar test.

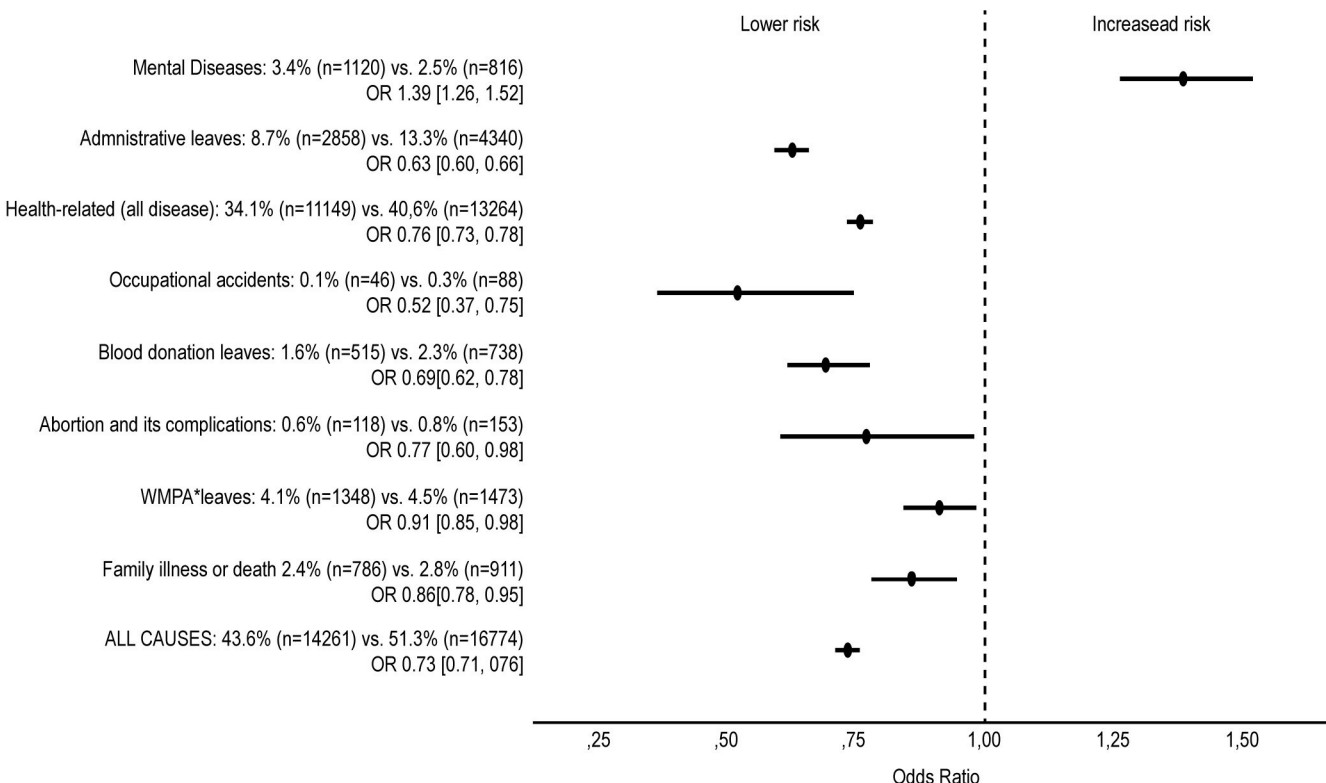

**Fig 1. Risk of absenteeism according to the classes of work leave during the pandemic period in comparison to the prepandemic period.** *WMPA—Wedding, Maternity, Paternity or Adoption leave. Statistical analysis was performed using generalized estimating equations (GEE) models based on the negative binomial distribution with a log-link function (unstructured correlation matrix).

## Discussion

Contrary to what was expected, our study showed a reduction in the number of work absences for all causes not directly related to SARS-CoV-2 infection during the pandemic period (Table 1), except for absences due to mental illness (Fig 1). Our study detailed the specific classes of work absence, not directly attributed to infection by SARS-CoV-2: health-related, administrative, marriage, maternity, paternity or adoption, illness or death of family members, blood donation, abortion, and occupational accidents. Despite reducing the absolute number, the proportion of classes of work absence in the two observation periods remained the same.

We speculate that these unexpected findings (reduction in work absences in 2020), could be attributed to the direction of health professionals to cope with the pandemic in such adverse and threatening circumstances and to the reduction of elective care due to the lockdown strategy. However, it can also be assumed that the temporary hiring of more employees, the improvement of the work process, and optimizing personal protective equipment may have contributed to reducing work absences during the pandemic period. A study that mapped the absenteeism of doctors in eight departments of a hospital in London during the first wave of COVID-19 in 2020 points to the importance of planning strategies to support health workers and points to alignment in the use of personal protective equipment as one of the key points [14].

In this context, these professionals were subjected to extreme working conditions, in addition to having to make difficult decisions to balance the needs of patients with their own physical and emotional needs [15–17], which may explain, at least in part, the increase in work

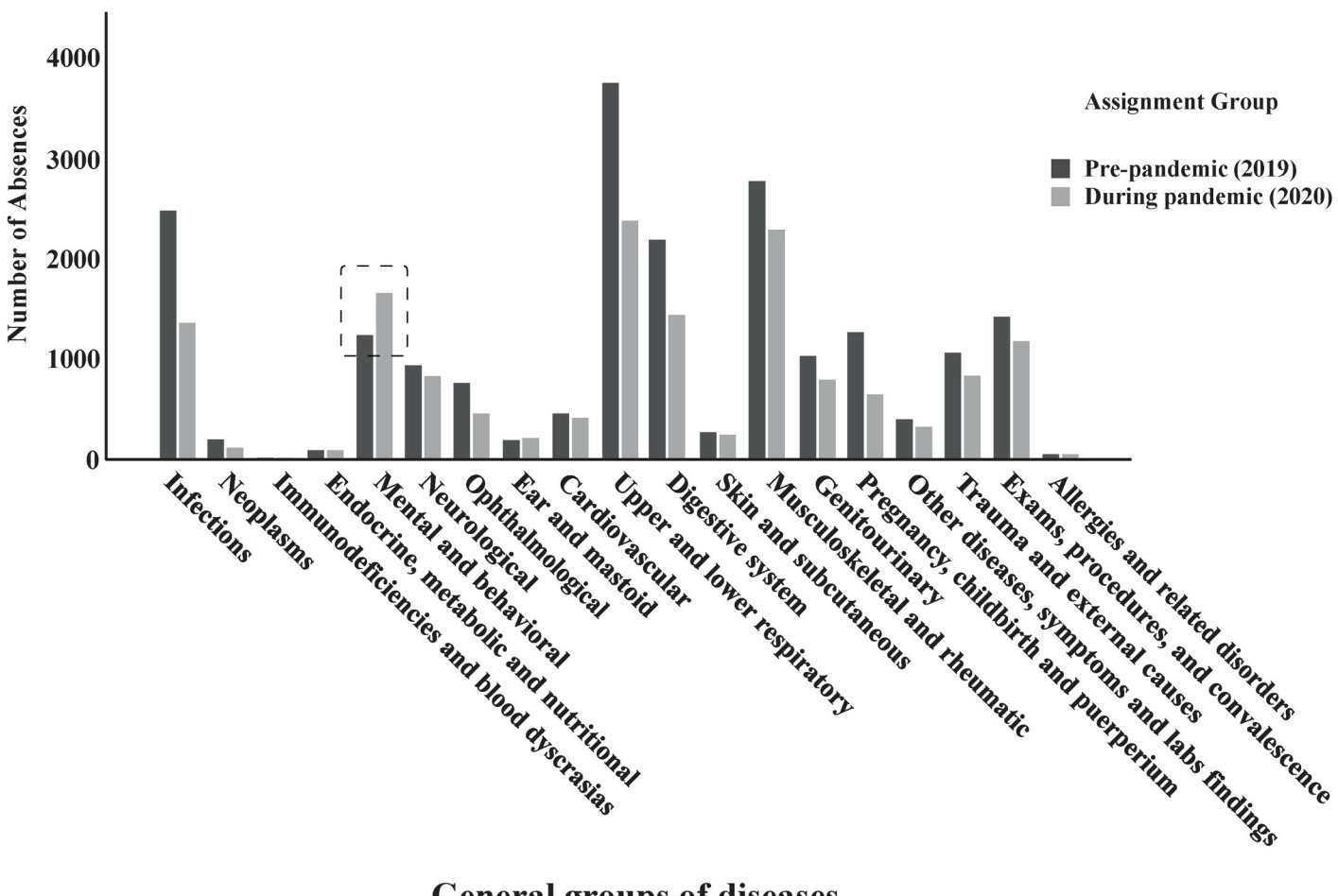

**General groups of diseases**

**Fig 2. Bar graph showing the absolute reduction in events of work absenteeism relating to health, except for mental illnesses, over the two periods of interest.**

absence due to mental illness. In line with our findings for mental illness, a study conducted in China between January and February 2020 involving 1257 frontline professionals showed that health professionals were at a higher risk of developing depression, anxiety, insomnia, and anguish, especially nurses and women [2]. Another survey of nurses in China identified the psychological needs for self-care regarding their health, safety and interpersonal relationships as fundamental for working during the pandemic, corroborating our mental health implications [18].

In our study, the risk of sickness due to mental illness in the pandemic period was 39% higher. In the literature, mental illnesses occur in 18 to 57% of health professionals who face outbreaks and epidemics [19, 20]. In a systematic review study regarding the psychiatric effects in health professionals during COVID-19, eight articles reported increases in symptoms of depression, anxiety, posttraumatic stress and sleep disorders [21]. Another systematic review that included the SARS, MERS, Ebola, influenza A and COVID-19 epidemics found a prevalence of psychiatric symptoms (17.3% to 75.3%), posttraumatic stress disorder (10–40%), depression (27.5 to 50.7%), insomnia (33–34.1%) and anxiety (45%) [22].

In this regard, in the present study, men had a lower risk of withdrawal compared to women, for whom the rate did not change due to exposure or the pandemic. As such, there

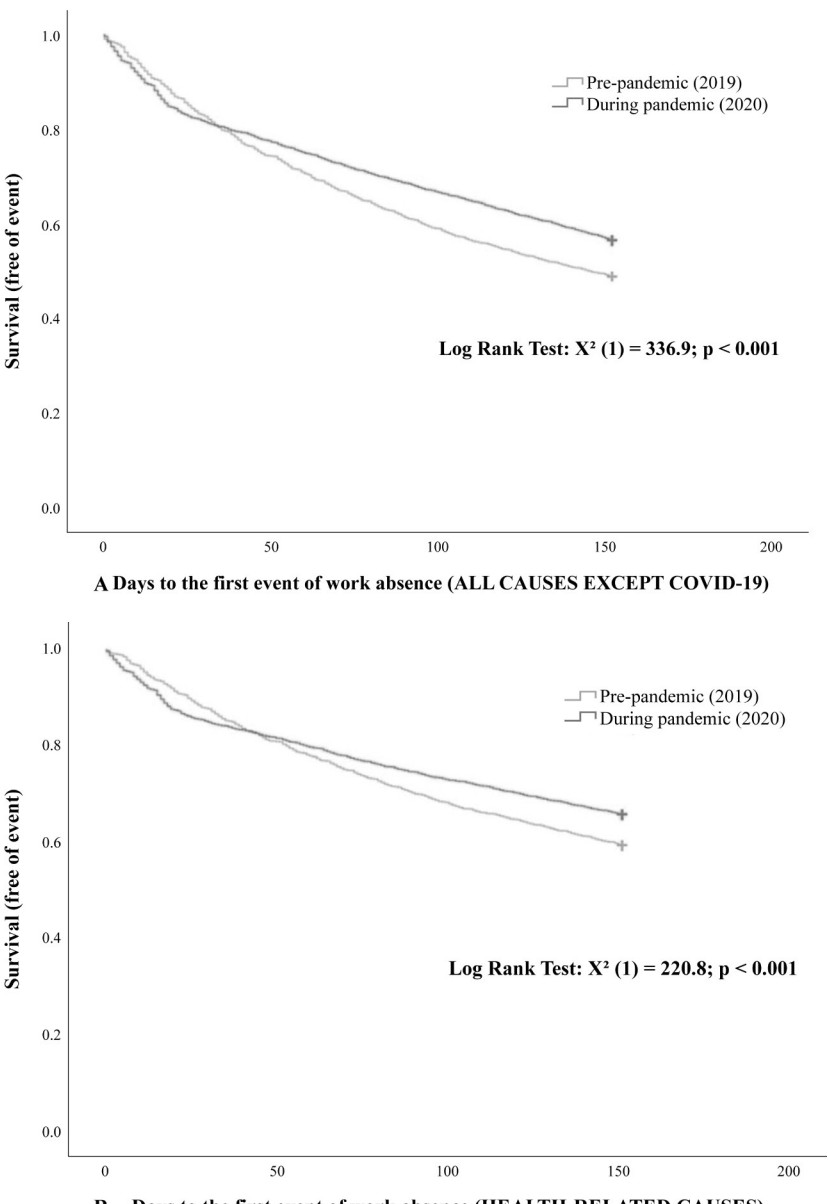

**Fig 3.** Survival curves for work absences in the pre- and pandemic periods for all causes (except for COVID-19 cases) (3A) and health-related causes (3B). Statistical analysis was performed using Kaplan-Meier curves and log-rank tests.

was more absenteeism among females. A similar result was highlighted in a systematic review that observed that females were more prone to mental illness-related absence [21].

Psychological support strategies have been suggested to reduce mental health impacts on health professionals [8, 19, 20, 23]. Training to improve resilience, psychological support groups, hotlines for psychological support, relaxation sessions and exercises have been described as strategies to mitigate the impact of mental illness in several countries and offer support to health professionals, optimizing their work capacity [12, 22]. A systematic review that evaluated interventions to improve resilience and psychological support concluded that the lack of knowledge of frontline professional needs, together with the lack of strategies and psychological skills of managers, are factors that hinder the support of health professional

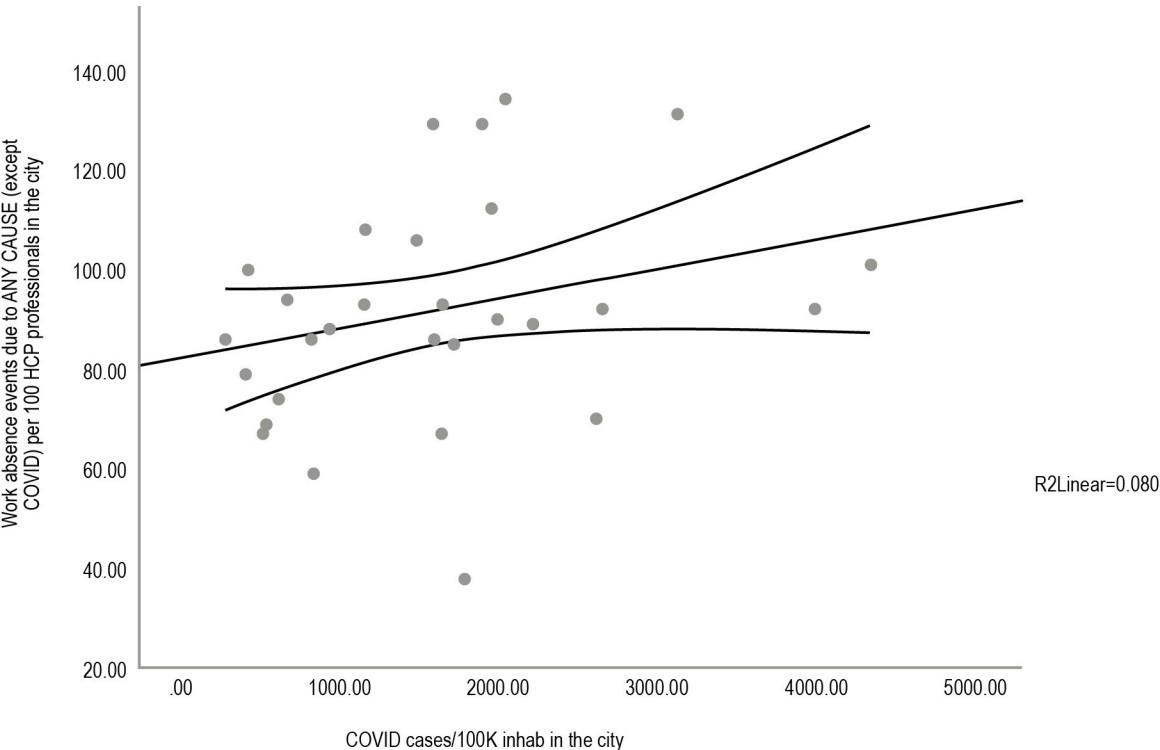

**Fig 4. Correlation between health professional absenteeism due to any cause (except COVID-19) and the cumulative prevalence of COVID-19 cases in university hospital cities.** HC = health care. Statistical analysis was performed using Spearman's rho test.

mental health. Facilitating factors were the implementation of psychological strategies by managers that were adaptable to local realities, effective communication, promoting a learning environment and professional enhancement [24].

Concerning our Kaplain-Meier curve observations, the initial panic generated by the unknown may have contributed to the initial increase observed in work absences due to all causes (except COVID-19) (Fig 3A) and health-related absences (Fig 3B). As knowledge about COVID-19 increased, there was an improvement in work processes, and possibly, with the hiring of more health professionals, work absences decreased over the observation period. A study published in July 2020, which compared absenteeism due to acute respiratory infection by military firefighters in Minas Gerais, Brazil, reported similar results in 2019 and 2020 (during the pandemic). The study showed that in February and March 2020, there was an initial increase in sick leave, followed by a reversal of this trend in April and May 2020, with a 2.4-fold reduction in the percentage of days not worked from May 2020. We must consider that firefighters are workers who are also at the forefront of combating the pandemic; thus, the results corroborate our findings [25].

We showed a correlation between the increase in work absence of health professionals for any cause (except for SARS-CoV-2 infection) and the increase COVID-19 cases per 100,000 population (Fig 4). A study published in September 2020 considered the installed assistance capacity of each state as one of the factors for the severity of COVID-19 in Brazilian states. In this regard, the absenteeism rate among professionals directly interferes with the care capacity [26].

Although the study focused primarily on the assessment of absenteeism for causes not directly related to COVID-19, we were also able to observe that 33.6% of the health workforce

were absent from work due to the suspicion or confirmation of COVID-19. This result differs from findings in other countries, such as Italy, in which the initial data pointed to a lower involvement (up to 20%) of the health workforce [27]; in the United States, also in April 2020, there was a reduction in the number of health professionals because of COVID-19, on the order of 3 to 11% [28, 29]. Possible explanations for the divergences in our findings include the initial lack of knowledge about the epidemiological characteristics of the pathogen, its high transmissibility, the need for prolonged direct contact with infected patients and the relative scarcity of personal protective equipment (PPE) [8, 27, 30].

The strengths of this study include the evaluation of a nationwide database, the large number of participants and the inclusion of professionals who work in providing health care directly or indirectly, and the ability of the data to reflect the heterogeneity of a country with continental dimensions and high levels social inequality. The results presented can be useful for planning and strategic management, to support the needs of health professionals directly or indirectly involved in combating the COVID-19 pandemic, which is still ongoing, as well as in possible similar situations in the future.

Recognizing these consequences requires health system managers to plan and intervene as quickly as possible to provide effective psychological support and treatment. It is also necessary to implement strategies that improve working conditions and minimize the detrimental effects on these professionals. Reducing the risk of illness in this workforce is so essential to society, especially when facing complex and unpredictable situations such as those experienced recently.

## Limitations of this study

Regarding study limitations, we were unable to clearly differentiate of absenteeism effectively related to mental health from that related to the fear of contracting the disease. To reduce this bias, data from official absences approved by an occupational medicine service were considered. Our analyses did not specify the most prevalent mental illnesses in the sample, which is a potential topic for future research. The fact that the study population came from university hospitals, environments with an academic purpose, and not primary care hospitals, may introduce bias concerning mental illness. Assistance hospitals of the same size as university hospitals, suffer more assistance pressure, fewer diagnostic resources and fewer qualified personnel. Thus, it is possible to infer that the incidence of mental illness in health care hospitals may be even higher than that observed in our study.

## Conclusion

Our study found a reduction in the total number of work absences unrelated to COVID-19 infection compared to the same prepandemic period. Despite this reduction, the number of work absences due to mental illness has increased.

These unexpected results point out how doctors, nurses, nursing technicians, physiotherapists, and other health workers committed to the mission of operationalizing the fight against the pandemic were at risk of physical illness both due to COVID-19 and due to work overload. Many of these professionals still paid a high price in terms of mental illness.

## Supporting information

**S1 File.**
(PDF)

**S2 File.**
(PDF)

**S3 File.**
(PDF)

**S4 File.**
(PDF)

**S5 File.**
(PDF)

**S1 Data.**
(ZIP)

**S2 Data.**
(ZIP)

## Acknowledgments

We thank Rodrigo Barbosa at Empresa Brasileira de Serviços Hospitalares–EBSERH.

## Author Contributions

**Conceptualization:** Adriana Ferreira Barros-Areal, Cleandro Pires Albuquerque, Giuseppe Cesare Gatto, Licia Maria Henrique da Mota.

**Data curation:** Adriana Ferreira Barros-Areal, Cleandro Pires Albuquerque, Marta Pinheiro Lima, Claudia Siqueira Besch, Giuseppe Cesare Gatto, Licia Maria Henrique da Mota.

**Formal analysis:** Adriana Ferreira Barros-Areal, Cleandro Pires Albuquerque, Nayane Miranda Silva, Dayde Lane Mendonça da Silva, Marta Pinheiro Lima, Everton Nunes da Silva, Licia Maria Henrique da Mota.

**Funding acquisition:** Adriana Ferreira Barros-Areal, Nayane Miranda Silva, Rebeca da Nóbrega Lucena Pinho, Everton Nunes da Silva, Laila Salmen Espindola, Licia Maria Henrique da Mota.

**Investigation:** Adriana Ferreira Barros-Areal, Cleandro Pires Albuquerque.

**Methodology:** Adriana Ferreira Barros-Areal, Cleandro Pires Albuquerque, Andrea Pedrosa Ribeiro Alves Oliveira, Ciro Martins Gomes, Fernando Araujo Rodrigues de Oliveira, Patrícia Shu Kurizky, Ana Paula Monteiro Gomides Reis, Thais Ferreira Costa, Licia Maria Henrique da Mota.

**Project administration:** Adriana Ferreira Barros-Areal, Cleandro Pires Albuquerque, Fernando Araujo Rodrigues de Oliveira, Luciano Talma Ferreira, Licia Maria Henrique da Mota.

**Resources:** Adriana Ferreira Barros-Areal.

**Supervision:** Adriana Ferreira Barros-Areal, Cleandro Pires Albuquerque, Licia Maria Henrique da Mota.

**Validation:** Adriana Ferreira Barros-Areal, Cleandro Pires Albuquerque, Andrea Pedrosa Ribeiro Alves Oliveira, Dayde Lane Mendonça da Silva, Ciro Martins Gomes, Patrícia Shu Kurizky, Ana Paula Monteiro Gomides Reis, Luciano Talma Ferreira, Thais Ferreira Costa, Licia Maria Henrique da Mota.

**Visualization:** Adriana Ferreira Barros-Areal, Thais Ferreira Costa, Heidi Luise Schulte, Laila Salmen Espindola, Licia Maria Henrique da Mota.

**Writing – original draft:** Adriana Ferreira Barros-Areal, Cleandro Pires Albuquerque, Nayane Miranda Silva, Rebeca da Nóbrega Lucena Pinho, Rivadávio Fernandes Batista de Amorim, Everton Nunes da Silva, Heidi Luise Schulte, Laila Salmen Espindola, Licia Maria Henrique da Mota.

**Writing – review & editing:** Adriana Ferreira Barros-Areal, Cleandro Pires Albuquerque, Nayane Miranda Silva, Rebeca da Nóbrega Lucena Pinho, Rivadávio Fernandes Batista de Amorim, Thais Ferreira Costa, Everton Nunes da Silva, Heidi Luise Schulte, Laila Salmen Espindola, Licia Maria Henrique da Mota.

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
