## [Decision Letter · Decision Letter 0]

29 Oct 2021

PONE-D-21-27324Impact of COVID-19 on the mental health of public university hospital workers in Brazil: a cohort-based analysis of 32,691 workersPLOS ONE

Dear Dr. Mota,

Thank you for submitting your manuscript to PLOS ONE. After careful consideration, we feel that it has merit but does not fully meet PLOS ONE’s publication criteria as it currently stands. Therefore, we invite you to submit a revised version of the manuscript that addresses the points raised during the review process.

ACADEMIC EDITOR: please address the methodpgical and presentation issues raised by the reviewers./>==============================

We look forward to receiving your revised manuscript.

Kind regards,

Rosemary Frey

Academic Editor

PLOS ONE

Journal Requirements:

2. We note that Figure 1 in your submission contain map images which may be copyrighted. All PLOS content is published under the Creative Commons Attribution License (CC BY 4.0), which means that the manuscript, images, and Supporting Information files will be freely available online, and any third party is permitted to access, download, copy, distribute, and use these materials in any way, even commercially, with proper attribution. For these reasons, we cannot publish previously copyrighted maps or satellite images created using proprietary data, such as Google software (Google Maps, Street View, and Earth). For more information, see our copyright guidelines: http://journals.plos.org/plosone/s/licenses-and-copyright.

3. We note that you have stated that you will provide repository information for your data at acceptance. Should your manuscript be accepted for publication, we will hold it until you provide the relevant accession numbers or DOIs necessary to access your data. If you wish to make changes to your Data Availability statement, please describe these changes in your cover letter and we will update your Data Availability statement to reflect the information you provide

Reviewers' comments:

Reviewer's Responses to Questions

**Comments to the Author**

1. Is the manuscript technically sound, and do the data support the conclusions?

Reviewer #1: Yes

Reviewer #2: Yes

2. Has the statistical analysis been performed appropriately and rigorously? 

Reviewer #1: No

Reviewer #2: Yes

3. Have the authors made all data underlying the findings in their manuscript fully available?

Reviewer #1: Yes

Reviewer #2: Yes

4. Is the manuscript presented in an intelligible fashion and written in standard English?

Reviewer #1: Yes

Reviewer #2: Yes

5. Review Comments to the Author

Reviewer #1: Thanks for inviting me to review the paper titled “Impact of COVID-19 on the mental health of public university hospital workers in Brazil: a cohort-based analysis of 32,691 workers.”

Authors have conducted a meaningful study. There are some methodological issues and reporting of study findings. After going through the paper in its entirety, hereby sharing my comments on paper.

Abstract: Comprehensive

Main file

Introduction-

The sentence-“adding to work-related stress that prior to the pandemic was at its highest level since 2014.[2]” needs simplification.

Methodology-

Statistics: Though authors talk about using GEE and McNemar test, they have not be clearly shown in tables. I would suggest an opinion of biostatistician be taken for the same.

Results: Figure 2, please check the placement of favors pre-pandemic and pandemic. As per the figure, it seems that mental illness related leaves were higher in pre-pandemic period.

Figure 3 mentions log_rang in place of log-rank please correct it.

Tables: The tables has been represented as such it is produced by the SPSS. Authors need to present it properly.

Discussion: Discussion has been appropriate.

Conclusion: fine

Reference: Fine

Reviewer #2: Overall, this is a timely & important study, well-written, with an adequately-conceived design & analysis for the purposes. Should be published, with a few minor revisions. (see below)

//Abstract is too lengthy. Need to shorten/condense a bit.

79 COVID-19 pandemic. Many aspects relating to this topic remains sparse in the literature,

//remain

80 particularly in developing countries such as Brazil which has historically suffered from

//have

82 The objective of the present study was to investigate the indirect impact of COVID-19 on

83 the health system workforce by assessing absenteeism relating to all causes not directly

84 attributable to suspected or confirmed SARS-CoV-2 infection, with particular focus on

85 mental health-related absenteeism

//Not quite clear how one distinguishes genuine absenteeism due to mental illness, from the use of this as an 'excuse' due to fear of catching Covid.

111 donation; abortion and complications; occupational accidents. Leave due to suspicion or

112 confirmation of SARS-CoV-2 infection and conversion to teleworking were not counted

113 among the outcomes of interest in the study. In addition to data relating to leave

//Does not help with this problem...there is no clear way to differentiate...should include this in limitations section

142 100 healthcare professionals) was assessed by Spearman’s rho. p-values were deemed

// P

159 The average age of the workers was 39,2 years (SD 7,52). There was a predominance

160 of females 70,3% (n = 22,982). The sample was comprised of 82,5% health

161 professionals, 13,7% support professionals, while the remaining 3,8% with no

//Given that this will have an international audience, better to use the convention of '.' when citing decimals throughout the article. Otherwise (e.g. see line 160) there will be confusion for some readers, when mixing decimals and large whole numbers in the same sentence or paragraph--for those who use that convention in their cultures.

//The discussion and conclusion sections contain some speculations re causes of the results which have little or no support in the study's results.

//There needs to be a separate 'limitations of this study' section.

6. PLOS authors have the option to publish the peer review history of their article (what does this mean?). If published, this will include your full peer review and any attached files.

Reviewer #1: **Yes: **Dr. Snehil Gupta (M.D.)

Reviewer #2: No

---

## [Author Response · Author response to Decision Letter 0]

9 Dec 2021

Dear Academic Editor and Reviewers

First, we would like to thank the opportunity to evaluate our work as well as the suggestions sent by the reviewers. 

The following are the answers to the questions listed by the editors:

1. As for Figure 1, we chose to remove the figure.

2. As for the availability of the data, we inform you that the raw study data are available at Supporting Information.

3. As for the references, all were reviewed and all DOI were included as requested. There were changes in references 1, for greater ease of access to information and in reference 3, which was replaced. It was included reference 14, to support the argumentation.

The following are the answers to the questions listed by Reviewer 1:

“Statistics: Though authors talk about using GEE and McNemar test, they have not be clearly shown in tables. I would suggest an opinion of biostatistician be taken for the same.”

The data were analysed by a biostatistician. All bivariate (unadjusted) analyses comparing proportions between groups in the pre- and pandemic periods were performed using the McNemar test for repeated measurements, as indicated in the methodology. All multivariate (adjusted) analyses, because it is a design of repeated measures used GHG models, as indicated in the methodology. However, to make clearer the use of each method chosen in each case, we add the information in the notes of each figure. 

“Results: Figure 2, please check the placement of favors pre-pandemic and pandemic. As per the figure, it seems that mental illness related leaves were higher in pre-pandemic period.

Figure 3 mentions log_rang in place of log-rank please correct it.

Tables: The tables has been represented as such it is produced by the SPSS. Authors need to present it properly.”

The suggestions regarding the correction of Figure 2, we chose to adjust the figure, which became figure 1. The subtitles of the figure were adjusted, previously written: "Favors during pandemic 2020 / Favors pre pandemic 2019", after the correction was written: "lower risk / increased risk". A footnote was included, with the statistical method used.

The orientation regarding figure 3 has been corrected. Previously was "log-rang", after correction the term used is "log-rank". A footnote has been added, with the statistical method used.

Table 1 was redone in order to meet the reviewer's suggestions, contemplating the instructions of the journal.

In figure 4, a footnote was included, with the statistical method used.

The S1Fig was included as Figure 2 in the article. 

“Reference: many of the references missing doi. Please add them wherever indicated.”

All DOI were included in the references. 

The following are the answers to the questions listed by Reviewer 2:

“Abstract is too lengthy. Need to shorten/condense a bit.”

The abstract was condensed to achieve the suggestion. It was already in the number of characters allowed (268 words), but additional reduction was made now with 249 words, as requested by the reviewer. 

All spelling corrections were performed. 

“The discussion and conclusion sections contain some speculations re causes of the results which have little or no support in the study's results.”

The text was reformulated, and reference 14 was included in order to support argumentation.

“There needs to be a separate 'limitations of this study' section.”

This section was included in the article.

We hope that the adjustments will meet the suggested recommendations and look forward to a conclusive response as soon as possible. 

Respectfully 

Prof. Licia Mota

---

## [Decision Letter · Decision Letter 1]

24 Feb 2022

PONE-D-21-27324R1Impact of COVID-19 on the mental health of public university hospital workers in Brazil: a cohort-based analysis of 32,691 workersPLOS ONE

Dear Dr. Mota,

Thank you for submitting your manuscript to PLOS ONE. After careful consideration, we feel that it has merit but does not fully meet PLOS ONE’s publication criteria as it currently stands. Therefore, we invite you to submit a revised version of the manuscript that addresses the points raised during the review process.

 Please make minor grammatical corrections as per reviewer comments.

We look forward to receiving your revised manuscript.

Kind regards,

Rosemary Frey

Academic Editor

PLOS ONE

Journal Requirements:

Reviewers' comments:

Reviewer's Responses to Questions

**Comments to the Author**

1. If the authors have adequately addressed your comments raised in a previous round of review and you feel that this manuscript is now acceptable for publication, you may indicate that here to bypass the “Comments to the Author” section, enter your conflict of interest statement in the “Confidential to Editor” section, and submit your "Accept" recommendation.

Reviewer #1: (No Response)

Reviewer #3: All comments have been addressed

2. Is the manuscript technically sound, and do the data support the conclusions?

Reviewer #1: Yes

Reviewer #3: Yes

3. Has the statistical analysis been performed appropriately and rigorously? 

Reviewer #1: I Don't Know

Reviewer #3: Yes

4. Have the authors made all data underlying the findings in their manuscript fully available?

Reviewer #1: Yes

Reviewer #3: Yes

5. Is the manuscript presented in an intelligible fashion and written in standard English?

Reviewer #1: Yes

Reviewer #3: No

6. Review Comments to the Author

Reviewer #1: Dear Authors, you have addressed the queries raised by me. However, if am not sure of the utility of the McNemar test here. I think chi-square test would be enough to support your finding. Similarly, the supporting information should be cited in the manuscript for readers to better comprehend that.

Reviewer #3: The author should correct grammatical errors and improve the writing.

Recommendation: Minor revision.

I do not have any potential conflict of interest to disclose.

7. PLOS authors have the option to publish the peer review history of their article (what does this mean?). If published, this will include your full peer review and any attached files.

Reviewer #1: **Yes: **Dr. Snehil Gupta, M.D., Assistant Professor, Dept of Psychiatry, AIIMS, Bhopal-462020

Reviewer #3: **Yes: **Silvio A. Ñamendys-Silva, MD, MSc, FCCP, FCCM

---

## [Author Response · Author response to Decision Letter 1]

10 Apr 2022

Dear Editor and Reviewers,

About the comments listed by the reviewer 1:

Regarding the commentary about the McNemar test, we agree that the (categorical) nature of the variables would give consideration to a Chi-squared test. However, our research followed a repeated-measures design, meaning that every participant were assessed twice (before and after).

Because of that, a regular Pearson's Chi-squared test for independent samples would not apply. Instead, we used the McNemar test, which is one of the most used approaches to compare correlated proportions (in paired samples). [1] The McNemar test does follow a Chi-squared distribution with one degree of freedom (in 2x2 table), but focuses in the discordances between the repeated measures.

The support information is quoted in the manuscript in the Data sharing item.

Answering the comments listed by the reviewer 3:

Regarding grammatical correction and writing in English, a new revision was performed to make the text appropriate to the required standards. We have included the certificate of review and professional publishing attached.

Regarding the changes requested to comply to Plos One's submission guidelines, all references to funding were removed from the manuscript and the ethical statement was inserted at the beginning of the methods section.

We hope that the adjustments will meet the suggested recommendations and look forward to a conclusive response as soon as possible.

Respectfully 

Prof. Licia Mota

---

## [Decision Letter · Decision Letter 2]

19 May 2022

Impact of COVID-19 on the mental health of public university hospital workers in Brazil: a cohort-based analysis of 32,691 workers

PONE-D-21-27324R2

Dear Dr. Mota,

We’re pleased to inform you that your manuscript has been judged scientifically suitable for publication and will be formally accepted for publication once it meets all outstanding technical requirements.

Kind regards,

Rosemary Frey

Academic Editor

PLOS ONE

Additional Editor Comments (optional):

Reviewers' comments:

Reviewer's Responses to Questions

**Comments to the Author**

1. If the authors have adequately addressed your comments raised in a previous round of review and you feel that this manuscript is now acceptable for publication, you may indicate that here to bypass the “Comments to the Author” section, enter your conflict of interest statement in the “Confidential to Editor” section, and submit your "Accept" recommendation.

Reviewer #1: All comments have been addressed

Reviewer #3: (No Response)

2. Is the manuscript technically sound, and do the data support the conclusions?

Reviewer #1: Yes

Reviewer #3: Yes

3. Has the statistical analysis been performed appropriately and rigorously? 

Reviewer #1: Yes

Reviewer #3: Yes

4. Have the authors made all data underlying the findings in their manuscript fully available?

Reviewer #1: Yes

Reviewer #3: Yes

5. Is the manuscript presented in an intelligible fashion and written in standard English?

Reviewer #1: Yes

Reviewer #3: Yes

6. Review Comments to the Author

Reviewer #1: Comments have been addressed, the article in its current form can be published. This is an important article from public health perspective. Large Data from Brazil would help policymakers to make appropriate COVID-19 related measures for the general public and health care professionals, thanks.

Reviewer #3: Dear Dr. Rosemary Frey

Academic Editor

I appreciate the opportunity to provide a review of the manuscript “Impact of COVID-19 on the mental health of public university hospital workers in Brazil: a cohort-based analysis of 32,691 workers” (PONE-D-21-27324R2).

Recommendation: Accept.

I do not have any potential conflict of interest to disclose.

Best Regards.

Silvio A. Ñamendys-Silva, MD,MSc, FCCP, FCCM

Internal Medicine & Critical Care Medicine

http://orcid.org/0000-0003-3862-169X

7. PLOS authors have the option to publish the peer review history of their article (what does this mean?). If published, this will include your full peer review and any attached files.

Reviewer #1: **Yes: **Dr. Snehil Gupta

Reviewer #3: **Yes: **Silvio A. Ñamendys-Silva, MD, MSc, FCCP, FCCM

---

## [Editor Report · Acceptance letter]

7 Jun 2022

PONE-D-21-27324R2 

Impact of COVID-19 on the mental health of public university hospital workers in Brazil: a cohort-based analysis of 32,691 workers 

Dear Dr. Mota:

I'm pleased to inform you that your manuscript has been deemed suitable for publication in PLOS ONE. Congratulations! Your manuscript is now with our production department. 

Kind regards, 

on behalf of

Dr. Rosemary Frey 

Academic Editor

PLOS ONE